# Spin-orbit driven superconducting proximity effects in Pt/Nb thin films

Machiel Flokstra [1], Rhea Stewart[1,2], Chi-Ming Yim [1,3], Christopher Trainer[1], Peter Wahl[1], David Miller [4], Nathan Satchell [5], Gavin Burnell [5], Hubertus Luetkens [6], Thomas Prokscha [6], Andreas Suter [6], Elvezio Morenzoni[6], Irina V. Bobkova[7,8,9], Alexander M. Bobkov[7] & Stephen Lee [1] ✉

Manipulating the spin state of thin layers of superconducting material is a promising route to generate dissipationless spin currents in spintronic devices. Approaches typically focus on using thin ferromagnetic elements to perturb the spin state of the superconducting condensate to create spin-triplet correlations. We have investigated simple structures that generate spin-triplet correlations without using ferromagnetic elements. Scanning tunneling spectroscopy and muon-spin rotation are used to probe the local electronic and magnetic properties of our hybrid structures, demonstrating a paramagnetic contribution to the magnetization that partially cancels the Meissner screening. This spin-orbit generated magnetization is shown to derive from the spin of the equal-spin pairs rather than from their orbital motion and is an important development in the field of superconducting spintronics.

When nonsuperconducting materials are joined to a superconductor to form hybrid structures, superconducting correlations may leak into the adjoining materials and a variety of interesting proximity phenomena can occur, leading to unique functionality[1,2]. Much work has focused on the junction between superconducting (S) and ferromagnetic (F) layers whereby the ferromagnet induces spin-triplet correlations by lifting the spin-degeneracy of the superconducting state[3,4]. For a homogeneous ferromagnet this leads to the generation of triplet spin pairs with only zero spin projection, while the creation of equal spin pairs requires the presence of magnetic inhomogeneity. These equal spin triplet pairs are highly tolerant of the ferromagnetic exchange field and due to their ability to propagate through relatively long pathways in ferromagnetic materials are referred to as long-ranged triplets. Their existence has been well-demonstrated experimentally[5-7]. In the diffusive limit, triplet correlations generated in this way possess the exotic property of having wavefunctions that are antisymmetric in their time component (they are of

odd-frequency[8]), leading to unusual properties such as the paramagnetic Meissner effect[9-11].

Recently there has been much consideration of the role of spin-orbit (SO) coupling in the creation and manipulation of spin-triplet correlations in hybrid structures[12-16]. In refs. 12,13 the general condition for the generation of long-range triplet correlations was considered. In the diffusive limit for superconducting proximity, a homogeneous ferromagnet in combination with a source of SO interactions can generate equal-spin triplets leading to long-range triplet currents[12,13]. SO coupling also lifts spin degeneracy and it has long been known theoretically that for a homogeneous superconductor in the presence of SO interactions the pair wavefunction can be a mixture of s-wave singlets and p-wave triplets[17]. The question thus arises of whether SO coupling, even in the absence of ferromagnetism, can generate triplet pair correlations in proximity coupled systems. Using the quasi-classical approximation usually employed for these systems, Bergeret and Tokatly [12,13] concluded that SO coupling by itself cannot induce

[1]School of Physics and Astronomy, SUPA, University of St. Andrews, St. Andrews, UK. [2]ISIS Neutron and Muon Facility, Rutherford Appleton Laboratory, Science and Technology Facilities Council, Didcot, UK. [3]Tsung-Dao Lee Institute, Shanghai Jiao Tong University, Shanghai, China. [4]School of Chemistry, University of St. Andrews, St. Andrews, UK. [5]School of Physics and Astronomy, University of Leeds, Leeds, UK. [6]Labor für Myonspinspektroskopie, Paul Scherrer Institut, Villigen PSI, Switzerland. [7]Institute of Solid State Physics, Chernogolovka, Russia. [8]Moscow Institute of Physics and Technology, Dolgoprudny, Russia. [9]National Research University Higher School of Economics, Moscow, Russia. ✉e-mail: sl10@st-andrews.ac.uk

triplet pairing. More recently a number of papers using other approaches have however suggested that triplet superconductivity can arise in Rashba SO metals and in metals with impurity-induced SO coupling, when coupled by proximity to a singlet superconductor[14,15,18–21]. These conclusions have also been supported recently by extension of the quasiclassical approach to include terms to first order in the SO interaction and appropriate modifications to the boundary condition between a superconductor and a high spin-orbit metal[16]. The results were also shown to be robust where the S layer itself also contains spin-orbit interactions. The triplet correlations generated were found to be long-ranged in that their decay in the normal metal is on the length scale of the normal state spin correlation length (and not shortened by the SO mechanism that creates them, in contrast to the case for short-ranged triplets inside a ferromagnetic material).

In this paper we use low-energy muon-spin rotation (LE-$\mu$SR) and scanning-tunneling spectroscopy (STS) to investigate the magnetic and electronic response of a Nb/Pt heterostructure. Both Pt and Nb are known to have strong SO interactions[22,23] (with Pt considerably stronger than Nb) and Pt is also a Stoner enhanced paramagnet[24]. We demonstrate the existence of a strong paramagnetic response of the superconducting condensate in the vicinity of the interface between the S and the spin orbit metal (SOM) layer, which has an unusual dependence on both temperature and magnetic field. Application of quasi-classical calculations[18,19] to model our precise sample layouts confirms these results and further demonstrates that the dominant mechanism is the spin associated with the triplet pairs rather than a modification of the Meissner currents. This is the first experimental demonstration of this novel effect and of the presence of odd-frequency triplet pair correlations in a proximity coupled superconductor in the absence of a strong exchange field.

## Results

The proximity induced superconductivity in the Pt layers was studied using STS. Figure 1A shows the normalized differential conductance at the surface of the Pt layer for two samples, Pt(10)/Nb(50)/Si (ST1) and Pt(2)/Nb(50)/Si (ST2) (numbers indicating the layer thickness in nm

and both samples were capped by 5 nm Au to perform the STS). For ST2 the gap across a 2 nm Pt layer is almost completely developed, whereas for ST1 the gap is reduced to around 10% of that in the Nb. This reflects the relatively short normal metal coherence length of around 10 nm inside the Pt (at our typical lowest measurement temperature for muon experiments of 2.5 K), which is very similar to the superconducting coherence length of our sputtered Nb itself. The important result here is that superconducting pairs can pass through a 10 nm thick Pt layer. Figure 1B shows the temperature dependence of the STS spectra for ST2, which shows good agreement with predictions obtained from the microscopic Usadel theory using realistic material parameters for the layout (see SI for more detail on the STS, including a direct comparison with the phenomenological Dynes model).

To further investigate the Pt/Nb interface we used LE-$\mu$SR to measure the local flux density $B(x)$ as function of depth into a Cu(40)/Pt(10)/Nb(50)/Si (SM2) sample (Fig. 2). In a sister sample Cu(40)/Nb(50)/Si (SM1) without a Pt spacer layer, the Cu layer is highly proximitized having a $\xi_s$ - 100 nm similar to that in Au[25], and due to the long electronic mean free path in Cu there is significant Meissner screening of the applied magnetic field $B_0 = 300$ G on both sides of the Cu/Nb interface (Fig. 2)[25]. In SM2, the presence of the 10 nm Pt layer between the Cu and Nb causes a significant reduction of the Meissner screening on both sides of the interface, though superconducting pairs are still able to cross the 10 nm Pt layer, as previously shown using the STS data. For all our presented muon measurements the typical error in the obtained flux density is of order 0.1 G. Using the quasiclassical Usadel equations (see SM for more detail on the quasiclassical modeling) we are able to produce a good description of the resulting Meissner profile $B(x)$ (Fig. 2) using only parameters obtained from independent LE-$\mu$SR, STS and transport measurements, and importantly, omitting any SO interactions. To compare the calculated profile with the muon results it is necessary to convolute $B(x)$ with the energy-dependent muon stopping profile, which, for SM2, yields the curve presented in Fig. 2 that gives a good description of the experimental muon results

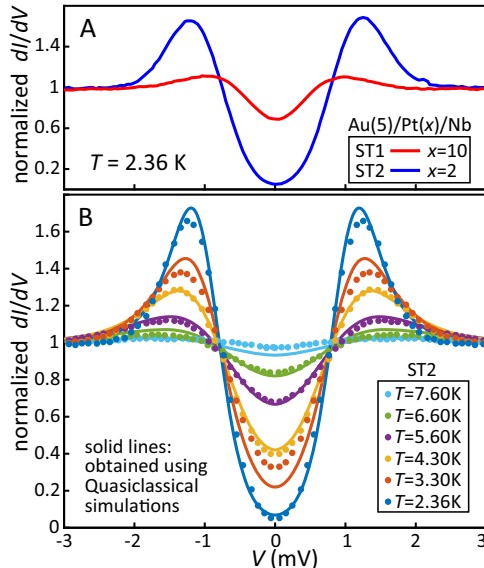

**Fig. 1 | Scanning tunneling spectroscopy data on Au(5)/Pt(x)/Nb(50)/Si(subs) (nm). A** The normalized differential conductance $\frac{dI}{dV}$ for $x = 2, 10$ nm. For $x = 2$ nm the Au layer is almost fully proximitized. The reduced gap for $x = 10$ nm suggest a coherence length $\xi_s$ - 10 nm. **B** Temperature evolution of the induced gap for $x = 2$ nm compared to predictions based on the microscopic Usadel theory (see supplementary materials for more detail).

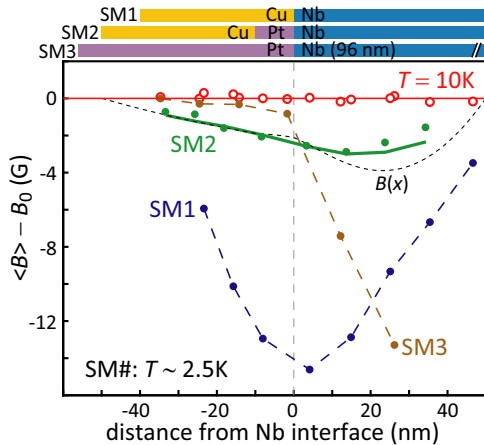

**Fig. 2 | Low-energy muon-spin rotation data for the average flux density $\langle B \rangle$ as function of average muon implantation depth $\langle x \rangle$ relative to the superconducting interface for Cu(40)/Nb(50) (SM1), Cu(40)/Pt(10)/Nb(50) (SM2) and Pt(56)/Nb(96) (SM3), with numbers indicating the layer thickness in nm.** Flux densities are presented relative to applied magnetic field $B_0$ (which is observed in all samples at $T = 10$ K with Nb in the normal state). In SM1, Cooper pairs diffuse far into the Cu due to its relatively long coherence length ($\xi_s$ - 100 nm) and flux screening develops well into the Cu. In SM2, the addition of the 10 nm thin Pt layer (which has a much shorter $\xi_s$ - 10 nm) shows a significant reduction of the flux screening. The SM2 data are modeled with a theoretical $B(x)$ (see text) which predicts the solid line for its corresponding $\langle B \rangle$ values. In SM3 the thicker Nb layer greatly enhances screening inside the Nb layer, however, the absence of screening near the Pt/Nb interface is unanticipated and is one of the key results presented in this paper. For all data points, error bars fit within the symbols used.

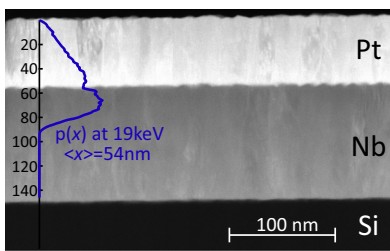

**Fig. 3 | Transmission electron microscopy (TEM) image of the Nb/Pt interface of SM3.** The overlay shows the stopping depth distribution of muons with an energy of 19 keV, corresponding to an average stopping depth of 54 nm ($x = -2$ nm in Fig. 2). At this energy roughly half of the muons come to rest in the Nb layer, probing its first 30−35 nm, yet the integrated flux expulsion is close to zero.

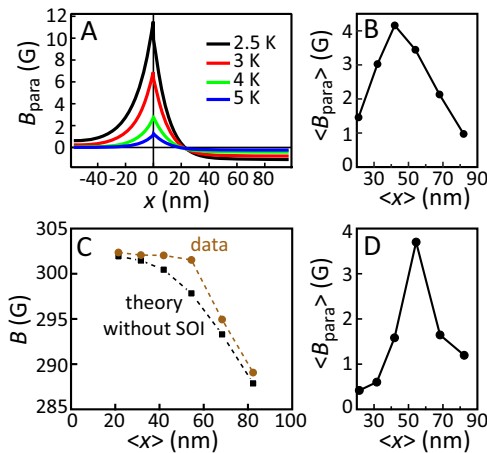

**Fig. 4 | Detailed analysis of the muon data of sample SM3 (see text). A** Calculated paramagnetic contribution to the response of the Pt/Nb, for various temperatures, using the full theoretical model including SO interactions and a spin Hall angle for the Pt of $\theta = 0.096$ (see supplementary materials for more information). **B** Convolution of the theory curve at $T = 2.5$ K **A** with the muon stopping profiles to obtain the calculated values for its contribution to the actual muon experiment. **C** The predicted measurement signal in the absence of SO interactions (squares) and the measurement data (filled circles). The difference between the two is shown in **D** and shows the (estimated) paramagnetic contribution to our measurement signal.

(see SM for more detail on the LEM technique and data analysis). The addition of the 10 nm thin Pt layer thus acts as something approaching a vacuum interface (i.e., greatly reducing proximity into the Cu and its contribution to the screening response). The resulting screening currents now flow mainly in the half space to the right of the Pt/Nb interface ($x > 0$).

The main results of this paper concern samples containing much thicker layers of Pt, such as Pt(56)/Nb(96)/Si (SM3) (Fig. 2). The choice of a thicker Nb layer, but still shorter than the magnetic penetration depth of about 150 nm[25], allows the possibility for larger screening currents to develop, amplifying subtle effects that scale with the size of the screening. The screening behavior of the Nb layer by itself and (in proximity to Cu and Au) is extremely predictable and has been previously measured in a variety of thin-film architectures[25–27]. As one might expect, for the higher muon energies probing inside the Nb layer a significantly larger screening is observed compared to SM2. However, the flux densities at each average implantation depth $\langle x \rangle$ have to be considered rather carefully in relation to the muon stopping profile used to sample them. It is instructive for sample SM3 to consider the point at -2 nm (e.g., 2 nm into the Pt measured from the Pt/Nb interface), which is almost exactly at the boundary between the Pt and Nb layers. The corresponding muon stopping profile is shown in Fig. 3, together with the TEM image of the interface. Due to scattering processes roughly half the muons for this measurement stop in the Nb layer, extending some 30 nm (around 3 coherence lengths) into the superconductor. A significant screening would thus be expected to be found at that muon energy. However, we find a much reduced value (see supplementary materials for similar results obtained on two related Pt/Nb samples with different layer thicknesses).

To model the SM3 data we make a detailed theoretical simulation for the specific sample layout, including extrinsic spin-orbit coupling in the dirty limit as discussed in refs. 18,19. More details of the application of this theory are presented in supplementary materials. We address the crucial question of how, in these particular structures, do the spin-triplet pairs contribute to the measured magnetic flux density? Two possible origins for additional paramagnetic contributions are identified: (1) due to the spin splitting of the quasiparticle density of states, caused by the spin-triplet pairs, and (2) due to (triplet) Meissner currents. This latter effect, which is typically the predominant one in F-S systems[11], is found in this purely SO coupled system to be orders of magnitude smaller than the first. The dominant effect is therefore one that has not been previously observed experimentally, a paramagnetic magnetization due to the local net spin-angular momentum of the condensate, as distinct from one arising from the orbital screening motion of the triplet pairs. In Fig. 4A we show this theoretical prediction for the spatial variation of the SO induced paramagnetic contribution, calculated as a function of temperature. The paramagnetic contribution is strongest at the Pt/Nb interface with an amplitude that increases with decreasing

temperature. The paramagnetic contribution also propagates into both layers and decays over several tens of nm. To compare our detailed theory model with the experimental data, we convolute the theory curve at $T = 2.5$ K with the actual muon stopping profiles. This gives the theoretically predicted SO contribution to the experimental muon data. The result is shown in Fig. 4B. Essentially this shows the additional contribution to the measurement expected from the spin-triplet pairs (generated by the SO interactions) when compared to a purely spin-singlet signal measured in the absence of SO interactions. Figure 4C shows the experimental muon data (filled circles) alongside the theoretically predicted measurement data (squares) when SO interactions are omitted. The difference between these curves, shown in Fig. 4D, allows a good estimate of the experimentally determined additional paramagnetic contribution. Comparison with Fig. 4B shows that it is similar both in shape and magnitude, with an expected maximum between the third and fourth implantation depth.

In order to explore further the development of the paramagnetic contribution we examine the temperature dependence of the total magnetic screening close to the interface, which for a singlet superconductor below $T_c$ would be a monotonic dependence that increases with the size of the gap[25]. From our theoretical simulations we find this monotonic behavior from the diamagnetic contribution. However, an unexpected upturn is found at lower temperatures due to the paramagnetic contribution. Both results are shown in Fig. 5A for two muon implantation energies, one probing predominantly at the Pt/Nb interface (19 keV) and the other probing deeper inside the Nb layer (27 keV). The upturn found in the theory model is predominantly due to the spin-triplet density increasing for lower temperatures. However, the low-temperature susceptibility of the Pt, which was experimentally determined as $\chi = 0.104/(T + 0.652)$ (see SI), also plays an important part, as does, to a lesser extent, the temperature dependence of the coherence length inside the Pt. Fig. 5B, C shows the corresponding measurement result, where for an implantation energy of 19 keV we indeed find this upturn, albeit with a slightly smaller magnitude of the effect. We have also measured the low-temperature field dependence of the expulsion, which even at $\langle x \rangle = 82$ nm has an unusual $\langle B \rangle \propto H^{\frac{3}{2}}$ dependence as opposed the usual linear dependence[25], indicating an unconventional superconducting state (see Fig. 5D).

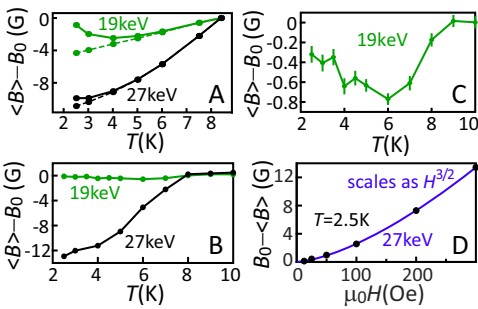

**Fig. 5 | The *T*-dependence of ⟨*B*⟩ for SM3. A** Theory calculation (including SO interactions and $\theta = 0.096$), showing the convolution of the total magnetic field with the muon stopping distribution at 19 keV (green solid) and 27 keV (black solid) implantation energies, corresponding to $\langle x \rangle = 54, 82$ nm, respectively. The dashed lines represent the results of the convolution of the diamagnetic response of the singlet correlations only (i.e., the difference between solid and dashed shows the paramagnetic contribution). **B** Experimental results at 19 keV and 27 keV implantation energies. **C** Expanded scale at 19 keV, showing the upturn close to the Pt/Nb interface as a result of the increasing paramagnetic contribution at lower temperatures. **D** Field dependence of the average flux expulsion measured at $T = 2.5$ K, 27 keV implantation energy, which follows a $H^{\frac{3}{2}}$ dependence as opposed to the usual linear Meissner relation.

## Discussion

The appearance of additional orbital paramagnetic contributions can arise in systems where odd-frequency spin-triplet correlations are generated when the spin degeneracy of the superconducting state is lifted. In an applied magnetic field the unusual time reversed response of the odd-frequency pairs can have a noticeable influence on the electromagnetic response of the condensate (see refs. [11],[28]) within a few coherence lengths of the interface, though deep within the S layer the singlet response will dominate. This has been observed using LE-$\mu$SR in a normal metal Au layer near the interface between S and F layer in Au/Ho/Nb[11] and in Au/$C_{60}$/Cu/Nb[28]. Nonetheless, in our present Nb/Pt system, however, we estimate these orbital contributions to be practically unresolvable. Recently the existence of spin triplets generated at Nb/Pt interfaces has been suggested via FMR experiments with correlations decaying over a similar length scale within the Nb to those observed here[29]. In our Pt/Nb system there is no ferromagnetic exchange field to convert spin-singlet pairs into spin-triplet pairs. However, by way of the (relatively strong) spin-orbit interaction in the Pt there is still a mechanism to create such spin-triplet pairs. These pairs are now in an environment devoid of a strong exchange energy, which would otherwise limit the coherence length of the opposite-spin spin-triplet pairs to around 1 nm. From our detailed theoretical modeling we find that the origin of the paramagnetic contribution arising near the interface with the superconductor is not from a paramagnetic Meissner current, as was the case for the Ho-based samples, but instead from the net spin of the spin-triplet pairs.

The theoretical expectation that a SOM proximity coupled to an S is capable of generating odd-frequency triplet correlations[14–16], coupled with the existence of triplet spin currents implied by the experiments of reference[29] (and intimated by refs. [30],[31]), provide a natural explanation of the anomalous screening behavior we observe.

We believe our results are distinct from previous reports of paramagnetic screening in that the origin of the contribution is from the spin of the triplet component rather than from their unusual orbital motion. The direct generation and manipulation of net spin correlation without the need for ferromagnetic elements is an important step forward in the direction to combine superconductivity and conventional spintronics.

## Methods

The samples were prepared by dc magnetron sputtering on Si (100) substrates at ambient temperature and a base pressure of $10^{-8}$ mbar.

Growth of all layers was performed at a typical Ar flow of 24 sccm and a pressure of 2–3 $\mu$bar with a typical growth rate of 0.2 nm s$^{-1}$. Growth rates for each material were calibrated by low angle X-ray reflectivity measurements on single material layers. The Nb target had a purity of 99.999% yielding sputtered Nb films with a typical superconducting transition temperature ($T_c$) of 8.7 K and a superconducting (Ginzburg-Landau) coherence length ($\xi_s$) of about 11 nm, determined from critical field measurements with field perpendicular to the sample plane. Two samples were grown for the STS measurements, Au(5)/Pt(10)/Nb(50)/Si (ST1) and Au(5)/Pt(2)/Nb(50)/Si (ST2), where for all samples thicknesses are given in nm and variations in the thickness of the cap layer are well below 1 nm. For the muon experiments two samples were grown to investigate the transmission characteristics of a thin Pt spacer later, Cu(40)/Nb(50)/Si (SM1) and Cu(40)/Pt(10)/Nb(50)/Si (SM2). A third sample was grown to study the influence of a thick Pt layer Pt(56)/Nb(96)/Si (SM3).

The quality of the interfaces between the Pt and Nb layers was investigated using transmission electron spectroscopy (TEM). The TEM measurements on the same sample show interfaces that are flat and clean both in terms of contrast (Fig. 3) and chemical specificity (see SI), with no evidence of interfacial alloying. In order to measure the density of states at the surface of the Pt layers we made use of two home-built scanning tunneling spectroscopy (STS) machines[32]. We used Pt-Ir tips, cut from a Pt wire and cleaned by in-situ field emission on a gold single crystal. All presented $\frac{dI}{dV}$ spectra are normalized with respect to the $\frac{dI}{dV}$ values at the region outside the superconducting gap where the spectral shape is completely flat. The local magnetic field as a function of depth into the sample was determined using low-energy muon-spin rotation (LE-$\mu$SR). SQUID magnetometry (M(T,H)) and standard transport techniques (R(T), Hc2(T)) were also used to test sample quality and extract material parameters.

## Data availability

The data that support the findings of this study are available from https://doi.org/10.17630/fa092d91-864d-4ae0-911e-7b630e6f0fe8.

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

## Acknowledgements

We acknowledge the support of the EPSRC through Grants Nos. EP/I031014/1 (P.W.), EP/J01060X (S.L.), EP/J010634/1 (G.B.), EP/L015110/1 (R.S.), EP/R031924/1 (P.W. and S.L.), EP/R023522/1 (S.L.), EP/S005005/1 (P.W. and C.Y.), EP/V028138/1 (G.B. and N.S.), and EP/L017008/1. This project has received funding from the European Union's Horizon 2020 research and innovation programme under the Marie Skłodowska-Curie grant agreement no. 743791 (SUPERSPIN) (N.S.). All muon experiments were undertaken courtesy of the Paul Scherrer Institut.

## Author contributions

N.S. and G.B. developed the samples; C.Y., C.T., and P.W. performed the STM measurements; M.F., R.S., S.L., H.L., T.P., A.S., and E.M. performed the muon measurements, in which H.L., T.P., A.S., and E.M. provided the beamline support; D.M. performed the SEM/TEM measurements; M.F., R.S., S.L., N.S., C.Y., and C.T. performed various support and characterization measurements; M.F. performed the quasiclassical modeling omitting spin-orbit interactions; I.B. and A.B. performed the detailed quasiclassical modeling including spin-orbit interactions; G.B. and P.W. helped designing the study; M.F., R.S., and S.L. conceived and designed the study, analyzed data and wrote the paper. All authors discussed the results and commented on the manuscript.

## Competing interests

The authors declare no competing interest.
