## [Peer Review File · Nature Communications]

REVIEWER COMMENTS

Reviewer #1 (Remarks to the Author):

I have read with great interest the manuscript by Flokstra and coworkers which reports on STM and low-energy muon spin rotation (LEM) measurements as well as theoretical modelling of the superconducting proximity effect in Pt/Nb multilayers with strong spin orbit coupling. Their main result is the observation of an anomalous paramagnetic signal that counteracts the Meissner screening and can be best described in terms of the magnetization of spin-triplet pairs that are induced via the SO coupling. This conclusion concerns a subject of great interest. However, it relies heavily on the comparison of the experimental data with the predictions of the quasiclassical calculations using dirty limit Usadel-equations.

The manuscript has a sufficiently detailed introduction that outlines a clear motivation of the work. The STS and LEM data appear to be of high quality and their analysis seems to be performed in a careful and meaningful manner.

The presentation of the central result, i.e. the observation of a strong paramagnetic signal in the vicinity of the Pt/Nb interface that cannot be explained without the magnetization of a spin-triplet induced by SO coupling, however, is not very clear and relies heavily on the comparison with the prediction of the theoretical models that are only discussed in the supporting material section. I find the discussion in the main part of the paper therefore rather abstract and difficult to follow.

There is also some potential confusion concerning the presentation of the magnetization profiles from the LEM data and the calculations with respect to the averaged implantation depth versus the actual depth into the sample. This concerns in particular Figure 4 where both notations are used in parallel. For example, what does it mean that the maximum of the paramagnetic signal in Fig. 4B occurs at a different position than the one in Fig. 4D?

A minor issue concerns the lower panel of Fig. 3 which does not seem to be mentioned in the main part of the manuscript (unless I overlooked it).

Overall, this makes me doubt whether the manuscript in its present form will be accessible to a broader readership that are not experts in this field. Therefore, I find it difficult to recommend this paper for publication in *Nature Communication*.

Reviewer #2 (Remarks to the Author):

The manuscript by Machiel Flokstra et al. reports an investigation of the local electron density of states and magnetic properties of heterostructures of the high spin-orbit coupling material Pt and the elemental superconductor Nb by means of scanning tunneling spectroscopy and muon-spin rotation measurements. The main observation is a paramagnetic contribution to the magnetization, which partially cancels the Meissner screening. The authors relate this effect to the presence of spin-triplet correlations.

This main conclusion of a spin-triplet contribution to the pairing mechanism induced by spin-orbit coupling rather than by ferromagnetism would be of sufficiently high significance, novelty and interest to warrant publication in a Journal with a broad readership like *Nature Communications*.

However, I have doubts whether all the conclusions of the manuscript are technically sound, as detailed below. Moreover, there is need for a strong improvement of the scholarly presentation of the Figures. Therefore, I cannot recommend publication of the manuscript in its current form. There is need for a major revision.

In the following, I will explain my criticism point by point.

1) The locally measured dI/dV spectra are analyzed and treated as an overall property of the heterostructure. However, as the STM image in Supplementary Figure S2 shows, which is unfortunately only taken for the Au(80)/Nb/Co sample, the grown heterostructures are very inhomogeneous. Moreover, the energies of the coherence peaks can locally vary, e.g. due to surface state scattering in the Au films. All this will surely result in a variation of the dI/dV spectra depending on the location on the surface. The general conclusions drawn from the dI/dV spectra are, therefore, highly questionable. As these conclusions are drawn from the Au(5)/Pt(x)/Nb/Si samples, the authors at least have to show according STM images of the surfaces of these heterostructures similar to those in Supplementary Figure S2, show how the dI/dV spectra vary as a function of location, and describe, whether they used any spatial averaging for the final data shown and analyzed in Fig. 1.

2) The STS data in Supplementary Figure S1 were measured for the Au/Nb/Co/Si system while the conclusion in the main manuscript are drawn from the Au(5)/Pt(x)/Nb/Si system. I expect a rather strong effect of the magnetic Co film on the superconductivity in the Nb layers. So I am wondering, how the STS results of these two strongly different heterostructures can be compared at all?

3) On page 2, right column, towards the end of the topmost paragraph, the authors conclude: "These results demonstrate that for Pt/Nb samples, there is no significant suppression of the superconducting gap." I am not sure, whether this conclusion is correct. Please note that, for proximitized heterostructures, the energy given by the position of the coherence peaks may deviate from the order parameter as seen, e.g., in the following theory work: Gábor Csire et al., *J. Phys.: Condens. Matter* 28, 495701 (2016).

4) The muon-spin rotation measurements have been done using samples with Cu capping layers instead of the Au capping used for the STS data. Why did the authors use different samples for the muon-spin rotation measurements than the ones used for the STS experiments? How can we be sure, that the results for two different sample systems can be compared one to one?

5) As seen in Fig. 3, there is a considerable amount of oxygen in the Nb film. So, it is rather a NbO film that has been grown, which may drastically change the superconducting properties.

6) Page 3, right column, top paragraph: the authors write, that "Fig. 4(B) shows the discrepancy between our experimentally observed values and our theory model when omitting SO interactions". However, without any experimental error bars, it is difficult to judge, whether the difference is significant. The authors have to add error bars.

In addition, the following minor points need to be improved.

7) Concerning all tunneling spectroscopy data: What means "normalized dI/dV "? The used normalization procedures need to be described in the methods section.

8) Also, which tip material/preparation have been used? This should be described in the methods section.

9) On page 2, left column on the top, the authors write: "Both Pt and Nb are known to have strong SO interactions and Pt is also a Stoner enhanced paramagnet." While this statement on the SOC and Stoner enhancement is definitely true for Pt, Nb is a lighter element such that the SOC is expected to be considerably weaker with respect to that of Pt. The authors should differentiate between the two elements and at least cite some literature on the SOC of the two elements and the Stoner enhancement of Pt.

10) On page 5, right column, top paragraph, the authors write “Two samples were grown for the STS measurements, Pt(10)/Nb(96)/Si (ST1) and Pt(2)/Nb(96)/Si (ST2)”. Please add the capping with Au layers.

11) The presentation style of the figures should be improved. In particular, strongly different font sizes are used for the different figures.

We thank the referees for their comments

REVIEWER COMMENTS

Reviewer #1

We thank the referee for raising the concern about the readability/presentation surrounding the central results (Fig 4) and have made appropriate changes throughout the manuscript to improve this. To answers the points of concern in detail:

1. The manuscript has a sufficiently detailed introduction that outlines a clear motivation of the work.
2. The STS and LEM data appear to be of high quality and their analysis seems to be performed in a careful and meaningful manner.
3. The presentation of the central result, i.e. the observation of a strong paramagnetic signal in the vicinity of the Pt/Nb interface that cannot be explained without the magnetization of a spin-triplet induced by SO coupling, however, is not very clear and relies heavily on the comparison with the prediction of the theoretical models that are only discussed in the supporting material section. I find the discussion in the main part of the paper therefore rather abstract and difficult to follow.
 - The theory used in our manuscript is an application of the quasi-classical calculations from Bergeret et al. (ref-16) and Huang et al (ref-17) to model our precise sample layouts. It predicts a paramagnetic response from spin-triplet pairing in the absence of ferromagnetic exchange fields and confirm our measurement results. It further demonstrates that in our samples the dominant mechanism is the spin associated with the triplet pairs rather than a modification of the Meissner currents. This is the first experimental demonstration of this novel effect and of the presence of odd-frequency triplet pair correlations in a proximity coupled superconductor in the absence of a strong exchange field. We have added this to the introduction part (page 2, first paragraph) and made further changes throughout the text to clarify the discussion.
4. There is also some potential confusion concerning the presentation of the magnetization profiles from the LEM data and the calculations with respect to the averaged implantation depth versus the actual depth into the sample. This concerns in particular Figure 4 where both notations are used in parallel. For example, what does it mean that the maximum of the paramagnetic signal in Fig. 4B occurs at a different position than the one in Fig. 4D?
 - We have improved the figure by first presenting the theoretical prediction (Fig4A,B) before presenting the LEM measurement in a way that best allows direct comparison with theory. We have also cleaned up the inconsistent/incorrect labelling of the x-axes. As for the position of the maximum of the peak heights. They are in fact not so different as they may appear due to the resolution of the implantation energies used (e.g. average implantation depths). For the theory prediction the peak lies somewhere between the 2nd and 4th energy point, while for the measurement the peak lies somewhere between 3rd and 5th energy point. There is thus a region of overlap (3rd to 4th energy point) where theory and experiment are in agreement.

5. A minor issue concerns the lower panel of Fig. 3 which does not seem to be mentioned in the main part of the manuscript (unless I overlooked it).
 - We removed the EDX image from the main paper (see also reviewer 2 comment 5), though a plot containing equivalent information remains in the SI.

Reviewer #2

We thank the referee for raising the concern about the presented STS data, as well as various minor points. To answers all the points of concern in detail:

1. The locally measured dI/dV spectra are analyzed and treated as an overall property of the heterostructure. However, as the STM image in Supplementary Figure S2 shows, which is unfortunately only taken for the Au(80)/Nb/Co sample, the grown heterostructures are very inhomogeneous. Moreover, the energies of the coherence peaks can locally vary, e.g. due to surface state scattering in the Au films. All this will surely result in a variation of the dI/dV spectra depending on the location on the surface. The general conclusions drawn from the dI/dV spectra are, therefore, highly questionable. As these conclusions are drawn from the Au(5)/Pt(x)/Nb/Si samples, the authors at least have to show according STM images of the surfaces of these heterostructures similar to those in Supplementary Figure S2, show how the dI/dV spectra vary as a function of location, and describe, whether they used any spatial averaging for the final data shown and analyzed in Fig. 1.
 - We have removed the erroneous comparison between the two different types of structures and removed the Au/Nb/Co results from the SI since they are not in any way necessary for our manuscript, can easily lead to confusion, and require additional data to support the statements we made about the comparison (see concern #2 as well). For completeness, we do nonetheless fully answer the referee's question/concern about these Au/Nb/Co system in the attached document "on_AuNbCo.pdf".
 - We have added to the manuscript the key finding of our STS results on Pt/Nb system: that superconducting pairs are able to penetrate a 10~nm thick Pt layer (page 2, right column, first paragraph).
 - We have added the requested STM data to the SI (fig S4).
2. The STS data in Supplementary Figure S1 were measured for the Au/Nb/Co/Si system while the conclusion in the main manuscript are drawn from the Au(5)/Pt(x)/Nb/Si system. I expect a rather strong effect of the magnetic Co film on the superconductivity in the Nb layers. So I am wondering, how the STS results of these two strongly different heterostructures can be compared at all?
 - This comparison has been removed from the paper as it was unnecessary and indeed not possible to conclude without additional data. For completeness, and to answer the referee, we have added the supporting data showing this comparison to the attached document "on_AuNbCo.pdf".
3. On page 2, right column, towards the end of the topmost paragraph, the authors conclude: "These results demonstrate that for Pt/Nb samples, there is no significant suppression of the superconducting gap." I am not sure, whether this conclusion is correct. Please note that, for proximitized heterostructures, the energy given by the position of the coherence peaks may deviate from the order parameter as seen, e.g., in the following theory work: Gábor Csire et al., J. Phys.: Condens. Matter 28, 495701 (2016).

- We have removed the comment as our statement was indeed ambiguous and unnecessary. The key result is simply that a (significant) gap structure is observed in the Au meaning that Cooper pairs are able to pass through the thin Pt layer.
4. The muon-spin rotation measurements have been done using samples with Cu capping layers instead of the Au capping used for the STS data. Why did the authors use different samples for the muon-spin rotation measurements than the ones used for the STS experiments? How can we be sure, that the results for two different sample systems can be compared one to one?
 - Au or Cu have both extensively been studied using LEM in a large variety of samples. For our type of experiments where we induce superconductivity into the normal metal, the only real difference between Au and Cu is that Cu has a much longer induced coherence length (see e.g. Phys. Rev. B 104, L060506 and references therein). This gives more room to develop screening currents in our thin film, generating a larger (and thus easier to measure) signal. For our muon measurements the Cu is thus favored, especially when the measurement signals are small. However, for STS we need a Au cap to get high quality scans.
 5. As seen in Fig. 3, there is a considerable amount of oxygen in the Nb film. So, it is rather a NbO film that has been grown, which may drastically change the superconducting properties.
 - The oxygen is distributed evenly throughout the Nb (as seen in the SI Fig.S11) and not of high enough concentration to form NbO, which would result in a rather different T_c . Our Nb is grown on our sputtering system using 5x9 purity targets and, over the course of well over a decade, have been extremely reproducible in a high quality Nb layer with a bulk T_c of about 8.7 and superconducting coherence length of around 10 nm. See e.g. citations 22-25 in the manuscript.
 6. Page 3, right column, top paragraph: the authors write, that "Fig. 4(B) shows the discrepancy between our experimentally observed values and our theory model when omitting SO interactions". However, without any experimental error bars, it is difficult to judge, whether the difference is significant. The authors have to add error bars.
 - For our LEM measurements the typical error in the average flux is of order 0.1 Gauss (see e.g. Fig.5C). In figures 3 and 4 this results in the error bars being fully obscured by the marker sizes of data points. Explicit information on the error bars is now added to the text (page 2, right-column, second paragraph).

In addition, the following minor points need to be improved.

7. Concerning all tunneling spectroscopy data: What means "normalized dI/dV "? The used normalization procedures need to be described in the methods section.
 - We normalized the spectra by the normal state differential conductance. This is now added to the methods section.
8. Also, which tip material/preparation have been used? This should be described in the methods section.
 - We used Pt-Ir tips, cut from a Pt wire and cleaned by in-situ field emission on a gold single crystal. This is now added to the methods section.
9. On page 2, left column on the top, the authors write: "Both Pt and Nb are known to have strong SO interactions and Pt is also a Stoner enhanced paramagnet." While this statement on the SOC and Stoner enhancement is definitely true for Pt, Nb is a lighter element such that the SOC is expected to be considerably weaker with respect to that of

Pt. The authors should differentiate between the two elements and at least cite some literature on the SOC of the two elements and the Stoner enhancement of Pt.

- We have added the information and relevant citations (refs 21-23) (page 2, first paragraph).

10. On page 5, right column, top paragraph, the authors write "Two samples were grown for the STS measurements, Pt(10)/Nb(96)/Si (ST1) and Pt(2)/Nb(96)/Si (ST2)". Please add the capping with Au layers.

- The information is added.

11. The presentation style of the figures should be improved. In particular, strongly different font sizes are used for the different figures.

- We have made the style and font sizes used in the figures more consistent.

REVIEWER COMMENTS

Reviewer #1 (Remarks to the Author):

The authors have responded well to the criticism and questions that I had raised in my previous report. The according changes to the revised manuscript have helped to make the main result of their work, i.e. the observation of spin-triplet Cooper-pairs that are induced by strong spin-orbit interaction, more easily accessible to the readership that is not specialized in this field. As such, I find the manuscript now suitable for publication in a journal like Nature Communications.

Reviewer #3 (Remarks to the Author):

The authors reply point par point to reviewer 2. Following this structure, I start commenting sequentially the "reply to reviewers". Secondly, I review "comment 1" and "comment 2" edited by the authors in the reply round. I finish with some other changes that should be performed.

Question 1.

The data requested are not included in the SI as indicated. Fig. S4 shows only a topography image and no STS data. In addition, the colorbar with min/max corrugation values for the topography is missing. A comment with respect the variation of the thickness of the cap layer should be included in the main text.

Question 4.

The authors argue that a change of the top cap layer is necessary for muon / STS measurements indicating that Au caps are needed to have high quality STM scans. I cannot agree on that. For Cu layers grow in epitaxial way, Cu surface do not represent any problem for STM measurements, see for example:

1. Crommie MF, Lutz CP, Eigler DM (8 October 1993). "Confinement of electrons to quantum corrals on a metal surface". *Science*. 262 (5131): 218-20
doi:10.1126/science.262.5131.218. PMID 17841867. S2CID 8160358.

Question 7.

Authors indicate that the normalization of STS data is performed with respect to the normal state. However, based on the document Comment 1, part 1; the choice of the value at which the normal conductance is defined is quite tricky. After normalization, one expects that flat part of the wings of the coherence peaks overlap, which is not the case in the data shown in the "comments". Authors should indicate more precisely at which energy the normal conductance has been obtained and why this value among other possibilities. In fact, this is crucial to see the evolution of the gap in a specific area, see comment 1.

Question 11.

The version I have received needs still improvement concerning the style and the font size.

Comment 1

Authors claim that the spatial variation of STS is not significant for systems with a layer thickness between 10 to 40nm. In order to claim so, the choice of normal conductance must be modified in the way the wings of the coherence peaks overlap. As the data are plot, we already observed that the value of the conductance at Fermi energy is not the same everywhere (for 10nm above 4K and for 40nm, this effect is already present at 2.5K). Moreover, I am afraid this behaviour will be enhanced once the normal state conductance for different locations overlap in the ± 10 meV range. This is an indication of superconducting gap modulation.

To claim uniform superconducting gap, authors should modify the normalization and plot a conductance map (2D maps of dI/dV at fixed energy).

Minor error. For comparison, it would be appreciated for the reviewer that curves obtained at the same temperature for different thickness have the same colour (e.g. R1 and R2 show inverted colors). In addition, this content should also be included in the SI.

Comment 2

Authors grow two systems: Au/Nb/Co/Si and Au/Pt/Nb/Si. I was wondering why the Nb/Co order has been reversed in Pt/Nb. For a direct comparison, Co and Pt should keep both the same layer order. I understand than Pt and Nb are chosen for the SOC, nevertheless the distance of the superconductor with respect to the STM tip and the path follow for the electrons along the layer material can have strong influence on the dI/dV measured.

Other changes:

Fig. S9- both axis label "?" must be modified.

REVIEWER COMMENTS

Reviewer #1 (Remarks to the Author):

The authors have responded well to the criticism and questions that I had raised in my previous report. The according changes to the revised manuscript have helped to make the main result of their work, i.e. the observation of spin-triplet Cooper-pairs that are induced by strong spin-orbit interaction, more easily accessible to the readership that is not specialized in this field. As such, I find the manuscript now suitable for publication in a journal like Nature Communications.

Reviewer #3 (Remarks to the Author):

The authors reply point par point to reviewer 2. Following this structure, I start commenting sequentially the "reply to reviewers". Secondly, I review "comment 1" and "comment 2" edited by the authors in the reply round. I finish with some other changes that should be performed.

We thank the reviewer for all the comments, especially on parts that had been previously removed from the manuscript prior to our resubmission, in response to the original comments of reviewer 2. We acknowledged that while interesting, these removed data on related systems were both unnecessary and distracting to the discussion and may indeed require further investigation. These further comments from reviewer 3 on those additional data will therefore be particularly helpful in preparing those future works for publication (e.g. comment 1 and comment 2).

Question 1.

The data requested are not included in the SI as indicated. Fig. S4 shows only a topography image and no STS data. In addition, the colorbar with min/max corrugation values for the topography is missing. A comment with respect the variation of the thickness of the cap layer should be included in the main text.

We should perhaps emphasize that the central result of our manuscript does not rely on any of the STS data. The main (and singular) purpose of the STS data is to show that cooper pairs are able to penetrate a 10~nm thick Pt layer, which is unambiguously confirmed by the superconducting gap structures seen in our STS data. The precise location of the coherence peaks, the depth of the gap, and the overall shape of the gap structure is of course interesting but not of immediate concern in relation to this confirmatory role. The detailed gap structure in this material is furthermore not something we would necessarily expect to be able to accurately describe since the presence of spin triplet pairs make for an unconvension gap structure. In figure 1 of the manuscript we included fits to the STS data obtained using the quasiclassical approach, and in the SI compare it to the Dynes model, but neither of these fits include SOI interactions. However, they do show that the lineshapes are reasonably well described (consistent with having a significant density of spin singlet pairing) and follow the predicted temperature dependence.

To reiterate the underlying concern about "Question 1" as stated by reviewer 2: "Moreover, the energies of the coherence peaks can locally vary, e.g. due to surface state scattering in the Au films. All this will surely result in a variation of the dI/dV spectra depending on the location on

the surface. The general conclusions drawn from the dI/dV spectra are, therefore, highly questionable.”

We agree with the reviewer that local variations of the gap can make it difficult to draw general conclusion from the dI/dV spectra when one is concerned about the precise lineshape and position of the coherence peaks. However, as discussed above, this is not an important factor for the data presented in our manuscript where the interest is primarily in whether we have conductance peaks at all or not, i.e. do we confirm electronically the significant penetration of Cooper pairs over this length scale as indicated magnetically by the muon measurements.

We thus argue that the requested STS data, to demonstrate the typical variations we observe in the position of the coherence peaks and the precise lineshapes, are a topic by themselves for a future publication but are not required for our current work.

We added a comment with respect to the variation of the thickness of the cap layer for the STS samples to the “Samples” section of the manuscript. (variations < 1nm, see e.g. the update figure S4 of the SI)

We added a colormap with min-max values to figure S4.

Question 4.

The authors argue that a change of the top cap layer is necessary for muon / STS measurements indicating that Au caps are needed to have high quality STM scans. I cannot agree on that. For Cu layers grow in epitaxial way, Cu surface do not represent any problem for STM measurements, see for example:

1. Crommie MF, Lutz CP, Eigler DM (8 October 1993). "Confinement of electrons to quantum corrals on a metal surface". *Science*. 262 (5131): 218-20
doi:10.1126/science.262.5131.218. PMID 17841867. S2CID 8160358.

In our correspondence to reviewer 2 it was not our intention to claim Cu can't be used as a capping layer for STS (as indeed highlighted by reviewer 3), though we can see how we might have given that impression. However, the intention was to state that for the specific case of Au and Cu capping layers grown on our sputtering machines, Au is favored to get the higher quality STS data, while Cu is favored to get the highest quality LEM data.

Question 7.

Authors indicate that the normalization of STS data is performed with respect to the normal state. However, based on the document Comment 1, part 1; the choice of the value at which the normal conductance is defined is quite tricky. After normalization, one expects that flat part of the wings of the coherence peaks overlap, which is not the case in the data shown in the “comments”. Authors should indicate more precisely at which energy the normal conductance has been obtained and why this value among other possibilities. In fact, this is crucial to see the evolution of the gap in a specific area, see comment 1.

The STS measurements shown in figures R1-R3 in the edited documents are on samples including the strong ferromagnet Cobalt and/or have much thicker gold capping layers. Several of these have indeed tricky lineshapes to normalize (but these are not samples used for the manuscript, which are all free from ferromagnetic elements). Figure R5 shows a typical STS

measurements on our sample ST2, Au(5)/Pt(2)/Nb(50), and this has a rather flat dI/dV for energies well outside the superconducting gap region, giving no particular problem with normalizing the data.

We added a line on the normalization to the "Methods" sections of the manuscript

Question 11.

The version I have received needs still improvement concerning the style and the font size.

> fixed.

Comment 1

Authors claim that the spatial variation of STS is not significant for systems with a layer thickness between 10 to 40nm. In order to claim so, the choice of normal conductance must be modified in the way the wings of the coherence peaks overlap. As the data are plot, we already observed that the value of the conductance at Fermi energy is not the same everywhere (for 10nm above 4K and for 40nm, this effect is already present at 2.5K). Moreover, I am afraid this behaviour will be enhanced once the normal state conductance for different locations overlap in the $\pm 10\text{meV}$ range. This is an indication of superconducting gap modulation.

To claim uniform superconducting gap, authors should modify the normalization and plot a conductance map (2D maps of dI/dV at fixed energy).

Minor error. For comparison, it would be appreciated for the reviewer that curves obtained at the same temperature for different thickness have the same colour (e.g. R1 and R2 show inverted colors). In addition, this content should also be included in the SI.

> We thank the reviewer for these comments which will be helpful in preparing a manuscript on these data. However, as discussed above, due to related remarks from reviewer 2, these data no longer formed part of the previously resubmitted manuscript nor of the SI.

Comment 2

Authors grow two systems: Au/Nb/Co/Si and Au/Pt/Nb/Si. I was wondering why the Nb/Co order has been reversed in Pt/Nb. For a direct comparison, Co and Pt should keep both the same layer order. I understand than Pt and Nb are chosen for the SOC, nevertheless the distance of the superconductor with respect to the STM tip and the path follow for the electrons along the layer material can have strong influence on the dI/dV measured.

> See comment 1. Again, these are issues that will be fully adressed in future publications where these data are included.

Other changes:

Fig. S9- both axis label "?" must be modified.

> fixed.

REVIEWERS' COMMENTS

Reviewer #4 (Remarks to the Author):

I was now asked in the third round of refereeing to comment on the response of the authors to the previous rounds, with special emphasis on the STS data. The initially presented data and also the first response were a bit confusing. The only conclusion from the STS spectra is the presence of a superconducting gap in a metal capping layer on top of the main sample consisting of a Nb bulk sample and a Pt layer of varying thickness. I believe that this fact is sufficiently supported by the STS data. The main conclusions are based on the LE muon data and their theoretical interpretation.

Minor comment: Fig 3 only has one panel, there is no "top" panel.

REVIEWER COMMENTS

Reviewer #4 (Remarks to the Author):

I was now asked in the third round of refereeing to comment on the response of the authors to the previous rounds, with special emphasis on the STS data. The initially presented data and also the first response were a bit confusing. The only conclusion from the STS spectra is the presence of a superconducting gap in a metal capping layer on top of the main sample consisting of a Nb bulk sample and a Pt layer of varying thickness. I believe that this fact is sufficiently supported by the STS data. The main conclusions are based on the LE muon data and their theoretical interpretation.

Minor comment: Fig 3 only has one panel, there is no "top" panel.

> fixed.